# Follicular Immune Landscaping Reveals a Distinct Profile of FOXP3^hi^CD4^hi^ T Cells in Treated Compared to Untreated HIV

**DOI:** 10.3390/vaccines12080912

**Published:** 2024-08-12

**Authors:** Spiros Georgakis, Michail Orfanakis, Cloe Brenna, Simon Burgermeister, Perla M. Del Rio Estrada, Mauricio González-Navarro, Fernanda Torres-Ruiz, Gustavo Reyes-Terán, Santiago Avila-Rios, Yara Andrea Luna-Villalobos, Oliver Y. Chén, Giuseppe Pantaleo, Richard A. Koup, Constantinos Petrovas

**Affiliations:** 1Department of Laboratory Medicine and Pathology, Institute of Pathology, Lausanne University Hospital, University of Lausanne, Rue du Bugnon 25, CH-1011 Lausanne, Switzerlandmichail.orfanakis@chuv.ch (M.O.);; 2Centro de Investigacion en Enfermedades Infecciosas, Instituto Nacional de Enfermedades Respiratorias “Ismael Cosio Villegas”, Mexico City 14080, Mexicogonavarr@gmail.com (M.G.-N.);; 3Department of Pathology and Laboratory Medicine, Emory University School of Medicine, Atlanta, GA 30322, USA; 4Institutos Nacionales de Salud y Hospitales de Alta Especialidad, Secretaría de Salud de México, Mexico City 14610, Mexico; 5Department of Laboratory Medicine and Pathology, Faculty of Biology and Medicine, Lausanne University Hospital, University of Lausanne, CH-1011 Lausanne, Switzerland; 6Service of Immunology and Allergy, Department of Medicine, Lausanne University Hospital, CH-1011 Lausanne, Switzerland; 7Vaccine Research Center, National Institute of Allergy and Infectious Diseases, National Institutes of Health, Bethesda, MD 20892, USA

**Keywords:** HIV, germinal center, imaging, CD4 T cells

## Abstract

Follicular helper CD4^hi^ T cells (T_FH_) are a major cellular pool for the maintenance of the HIV reservoir. Therefore, the delineation of the follicular (F)/germinal center (GC) immune landscape will significantly advance our understanding of HIV pathogenesis. We have applied multiplex confocal imaging, in combination with the relevant computational tools, to investigate F/GC in situ immune dynamics in viremic (vir-HIV), antiretroviral-treated (cART HIV) People Living With HIV (PLWH) and compare them to reactive, non-infected controls. Lymph nodes (LNs) from viremic and cART PLWH could be further grouped based on their T_FH_ cell densities in high-T_FH_ and low-T_FH_ subgroups. These subgroups were also characterized by different in situ distributions of PD1^hi^ T_FH_ cells. Furthermore, a significant accumulation of follicular FOXP3^hi^CD4^hi^ T cells, which were characterized by a low scattering in situ distribution profile and strongly correlated with the cell density of CD8^hi^ T cells, was found in the cART-HIV low-TFH group. An inverse correlation between plasma viral load and LN GrzB^hi^CD8^hi^ T and CD16^hi^CD15^lo^ cells was found. Our data reveal the complex GC immune landscaping in HIV infection and suggest that follicular FOXP3^hi^CD4^hi^ T cells could be negative regulators of T_FH_ cell prevalence in cART-HIV.

## 1. Introduction

Despite the intense research in HIV pathogenesis and cure approaches over the last 40 years, several aspects related to relevant immune cell and viral dynamics are still not well understood. Combination antiretroviral therapy (cART) has extended the life expectancy and improved the quality of life of People Living With HIV (PLWH). Despite the blocking of HIV replication, cART cannot eradicate the virus [1,2]. The latent infection of resting CD4^hi^ T cells is a main contributor to the lifelong persistence of HIV [3]. Integrated HIV DNA can be detected in blood and peripheral tissues, but recent studies suggest that genetically intact proviruses are mainly detected in the lymph nodes (LNs) [4]. Chronic HIV infection results in dramatic changes in LN architecture and the loss of stromal cells and adaptive immune cell responses [5]. Follicular helper CD4^hi^ T cells (T_FH_), a pivotal mediator of efficient B cell responses to pathogens [6,7], represent a major contributor to the HIV reservoir within the central memory CD4^hi^ T cell compartment [8,9]. Non-human primate (NHP) studies have shown an accumulation of T_FH_ cells in chronically SIV-infected compared to non-infected animals, a profile associated with an increased frequency of activated germinal center B cells and secretion of SIV-specific antibodies [10]. Progression to AIDS (acquired immunodeficiency syndrome, advanced stage of disease) was associated with an advanced loss of T_FH_ cells in SIV-infected NHPs [11]. In addition to T_FH_ and GC B cell altered dynamics, chronic HIV/SIV results in the increased cell density of LN/follicular effector CD8^hi^/GrzB^hi^CD8^hi^ T cells [12], deregulated immune-regulatory (T_REG_) and follicular immune-regulatory (T_FR_) CD4+ cells [13,14], the infiltration of inflammatory cell subsets and excessive fibrosis [5,15]. Foxp3^hi^CD4^hi^ T cells were reported to suppress the capacity of T_FH_ cells for proliferation and cytokine secretion, in ex vivo HIV-focused studies [14]. Additionally, the IL-10 and CTLA-4 expression of TFR cells were up-regulated in treatment-naïve PLWH [16]. Innate immune cells, which contribute to adaptive immune responses either by antigen presentation or by the secretion of immunomodulatory cytokines [17], play an important role in HIV pathogenesis [18,19], while their role in cure strategies [20] is a field under development. 

Several studies have focused on the characterization of LN-derived T cells, mainly using LN-derived cell suspensions and flow or mass cytometry-based assays [21] or single-cell RNA analysis [22,23]. However, these experimental approaches lack information about immune cell spatial organization. The in situ characterization of immune cell types, using multiplex imaging methodologies and appropriate cohorts of control and disease samples, can provide important insights into the prevalence, phenotype, and spatial organization of the relevant immune cells in HIV, which could also indicate possible mechanistic interactions between specific cell types.

Herein, we applied multiplex imaging combined with advanced computational tools for the comprehensive characterization of immune landscaping in viremic and cART HIV LNs compared to reactive control LNs or tonsils from non-HIV infected individuals. Our results indicate that both viremic and cART HIV LNs can be further grouped based on T_FH_ cell density. These groups exhibited distinct profiles of Foxp3^hi^CD4^hi^ T, Granzyme B (GrzB)^hi^CD8^hi^ T and innate immune cell subsets. Our work points to a possible role of FOXP3^hi^CD4^hi^ T cells as regulators of T_FH_ cells in HIV, particularly in cART individuals. Our neighboring analysis revealed a distinct distribution pattern for both T_FH_ and follicular (F-)FOXP3^hi^CD4^hi^ T between PLWH subgroups, as well as compared to non-infected donors.

## 2. Materials and Methods

### 2.1. Human Material

The tissue samples used in this study were obtained from (i) the Centro de Investigacion en Enfermedades Infecciosas (CIENI), Instituto Nacional de Enfermedades Respiratorias (INER) in Mexico City, Mexico (viremic LNs), (ii) the University of Washington, Seattle, WA, USA (cART HIV LNs) and (iii) the archives of the Institute of Pathology of Lausanne University Hospital, Switzerland (control LNs). Tonsillar tissues were obtained from anonymized children who underwent routine tonsillectomy at the Hospital de l’Enfance of Lausanne. All procedures were in accordance with the Declaration of Helsinki and approved by the appropriate Institutional Review Board/Ethical Committee: (i) all tissue samples from PWH were procured with explicit written informed consent from participants prior to donation, adhering strictly to the principles outlined in the Declaration of Helsinki.

### 2.2. Tissue Processing

Fresh tissues were fixed as soon as possible after biopsy for 16–24 h in formalin or 4% paraformaldehyde and processed for the preparation of formalin-fixed, paraffin-embedded (FFPE) blocks using standard procedures at the corresponding pathology departments. All downstream tissue processing was carried out in our laboratory. The blocks were sequentially cut into 4 μm sections and prepared on Superfrost glass slides (Thermo Scientific, Waltham, MA, USA, Ref. J1800AMNZ), dried overnight at 37 °C and stored at 4 °C. Before staining, the slides were heated on a metal hotplate (Stretching Table, MEDITE Medical GmbH, Burgdorf, Germany, OTS 40.2025, Ref. 9064740715) at 65 °C for 20 min. This melting step ensures the proper adherence and deparaffinization of the tissue section. Fluorescent multiplex immunohistochemistry (mIHC) staining was performed on the Ventana Discovery Ultra Autostainer from Roche Diagnostics (Ventana Medical Systems, Tucson, AZ, USA).

### 2.3. Confocal Imaging Assays

#### 2.3.1. Tissue Staining and Data Acquisition

Tissue sections were sequentially subjected to antibody blocking using Opal blocking/antibody diluent solution (ARD1001EA) staining with primary antibodies (details on antibodies, clones and dilutions are listed in Appendix A), incubation with secondary HRP-labeled antibodies for 16 min, detection with optimized fluorescent Opal tyramide signal amplification (TSA) dyes (Opal 7-color Auto-mation IHC kit, from Akoya (Marlborough, MA, USA), Ref. NEL821001KT) and repeated antibody denaturation cycles. The samples were then counterstained with Spectral DAPI from Akoya for 4 min, rinsed in soapy water and mounted using DAKO mounting medium (Dako/Agilent, Santa Clara, CA, USA, Ref. S302380-2).

Images were acquired using a Leica Stellaris 8 SP8 confocal system, equipped with Leica Application Suite X (LAS-X)-4.6.1.27508 software, at 512 × 512 pixel density and 0.75× optical zoom using a 20× objective (NA), unless otherwise stated. Frame averaging or summing was never used while obtaining the images. At least 70% of each section was imaged, to ensure an accurate representation and minimize selection bias. Tissues stained with a single antibody–fluorophore combination were used to create a compensation matrix via the Leica LAS-AF Channel Dye Separation module (Leica Microsystems, Wetzlar, Germany), which was used to correct fluorophore spillover (when present), as per the user’s manual.

#### 2.3.2. Quantitative Imaging Analysis (Histo-Cytometry)

A confocal image analysis was performed with Imaris software version 9.9.0 (Bitplane). Quantitative data were generated from the images through Histo-cytometry analysis [12,24], as previously reported. In brief, the Surface Creation module of Imaris was used to generate 3-dimensional segmented surfaces (based on the nuclear signal) of spillover-corrected images. Data generated from Histo-cytometry, such as average voxel intensities for all channels, in addition to the volume and sphericity of the 3-dimensional surfaces, were exported in Microsoft Excel format. The files were converted to comma separated value (.CVS) files, and the data were imported into FlowJo (version 10) to be further analyzed and quantitated. Well-defined areas devoid of background staining were included in the analysis, and the data were quantified either as relative frequencies or as cell counts normalized to the total follicular area screened. Optimal z-stack settings were applied in all collected images. Maximum Intensity Projections (MIPs) are presented throughout the manuscript.

#### 2.3.3. Data Analysis–Neighboring Analysis

The distance between the relevant cell subsets (CD20^hi^, PD1^hi^CD57^hi/lo^ etc.) was calculated with Python 3.10.9 using the SciPy library [25]. The matrix interaction was created using X and Y coordinates from each cell phenotype, and the median distance was extracted. Furthermore, to characterize the probability of observing different patterns of cellular distribution across Regions Of Interest (ROIs) and patients, we studied the curves generated from the Ripley’s G function and the theoretical Poisson curve using pointpats 2.3.0 (https://doi.org/10.5281/zenodo.7706219, accessed on 25 September 2023). The area between the empirical and theoretical Poisson curve was extracted using the NumPy library [26]. ROIs with at least 20 positive cells for each cell subset under investigation were analyzed. The data were presented as (i) bar graphs, showing the range of all the various distances measured (X axis) and the frequency or count of B cells that fell within each distance range and (ii) dot plots, where each dot represents the mean value of the minimum distances between two cell populations for each follicular area.

#### 2.3.4. Viral Load Measurement

The m2000 system (Abbott, Abbot Park, IL, USA) was used to perform an automated real-time polymerase chain reaction (PCR) for the determination of HIV plasma viral load (pVL), with a detection limit of 40 HIV RNA copies/mL. Flow cytometry with the AQUIOS Tetra-1 Panel in AQUIOS CL (Beckman Coulter Life Sciences, Indianapolis, IN, USA) was used to determine CD4^hi^ T cell counts. All PWLH involved in the current study were infected with clade B HIV.

### 2.4. Statistical Analysis

For the imaging data analysis, the Mann–Whitney test and simple linear regression analysis were used. The *p*-values of the Mann–Whitney test were corrected using the False Discovery Rate (FDR) correction test [27] with q = 0.05, for multiple comparisons (both uncorrected and corrected *p* values for each figure are shown in Appendix A). The analysis and graphs were generated using the GraphPad Prism 8.3.0 software. For statistical significance, a *p* value < 0.05 was considered.

## 3. Results

### 3.1. Similar Profiles of Follicular Helper CD4^hi^ T Cell Densities in Viremic and cART HIV LNs

T_FH_ cells are major contributors to HIV reservoir maintenance [28], as well as the development of broadly neutralizing antibodies [29]; we sought to investigate their in situ cell density in HIV-infected, compared to non-infected, tissues. We used tonsils and non-infected, cancer free, reactive LNs characterized by follicular hyperplasia as strict control groups (Appendix A) (Table 1). We started our analysis by employing a multiplex immunofluorescence imaging assay that allows for the simultaneous identification of major GC B and T cell subsets (Figure 1A). The gating strategy applied for the identification and quantitative analysis of relevant cell subsets by the Histo-cytometry pipeline [12,24] is shown (Figure 1B). The in situ density of CD20^hi/dim^ cells was used for the identification of individual follicular areas (highly enriched in CD20^hi/dim^ cells), as well as the ‘total follicular’ area (Figure 1B). As expected [5,30], PWLH tissues harbor both ‘preserved’ and ‘irregular’ follicular structures (Figure 1A and Appendix A). CD4^hi^ T cell subsets within the ‘total follicular’ area were analyzed based on their expression of PD1 and CD57 (Figure 1B). A preliminary analysis revealed low expression levels of CD4 in certain T_FH_ cells, particularly the ones expressing CD57 (Appendix A). Several studies have shown that T_FH_ cells express a unique PD1^hi^ phenotype [9,10,31] compared to other CD4^hi^ T cells, while PD1 expression per cell (judged by Mean Fluorescence Intensity) of follicular CD8^hi^ T cells is 4–5 times lower than that of T_FH_ [12]. Therefore, the expression level of PD1 can serve as an in situ T_FH_ identifier. To avoid inconsistencies and the misinterpretation of our data, especially for the CD57^hi^ T_FH_ cell subset, we chose to directly analyze the PD1^hi^CD57^hi/low^ cells in GCs (Figure 1B). The gating for setting the threshold of these biomarkers is shown in Appendix A. Their expression level in extrafollicular areas, as well as the manual inspection of their fluorescence intensities in the raw images, was used as a reference to set the cut-off for ‘high’ values. The backgating of PD1^hi^ and CD57^hi^ cells identified by Histo-cytometry (shown as spheres) to the original image showed a high concordance between the digitally identified cells and their original counterparts identified by immunofluorescence staining (Figure 1C).

The calculation of PD1^hi^ cell densities (normalized cell counts per mm^2^) allowed for the further grouping of HIV viremic and cART tissues into two subgroups characterized by significantly different cell densities of PD1^hi^ T_FH_ cells: one with high PD1^hi^ T_FH_ cell densities (hereafter ‘high-T_FH_’) and one with low PD1^hi^ T_FH_ cell densities (hereafter ‘low-T_FH_’) (Figure 1D, left panel). To assess the heterogeneous (or not) prevalence of PD1^hi^ T_FH_ cells across an individual tissue, the cell densities of PD1^hi^ T_FH_ cells per follicle for every tissue were analyzed. A great variability in PD1^hi^ T_FH_ cells was observed in all tissues, particularly in tonsils and control LNs (Figure 1D, right panel).

Next, the expression of CD57, a carbohydrate epitope that marks a T_FH_ cell subset with a distinct positioning and function in human LNs [32,33] was investigated. As expected [33], the PD1^hi^CD57^hi^ group represents a subset of T_FH_ cells (Figure 1E). PD1^hi^CD57^hi^ T_FH_ cells exhibited a similar cell density profile to PD1^hi^ cells (Figure 1E) and a heterogenous prevalence across the tissue (Appendix A). A strong correlation between PD1^hi^ and PD1^hi^CD57^hi^ T_FH_ cell densities was found for tonsil, control and cART LNs (Figure 1F). This association was less significant in the high-T_FH_ PLWH viremic subgroup (Figure 1F). Although not statistically significant, a negative association was observed between the PD1^hi^ T_FH_ cells and pVL in viremic PLWH, as well as a trend for a higher pVL in the low-T_FH_ compared to high-T_FH_ viremic PWLH subgroup (Figure 1G, lower and upper panel, respectively). Therefore, in agreement with our previous data for viremic SIV infection [10], two subgroups defined by significantly different cell densities of T_FH_ cells were identified in viremic as well as cART HIV LNs.

### 3.2. A Distinct Positioning Profile of T_FH_ Cells in HIV-Infected Compared to Non-Infected Tissues

Next, the spatial positioning of GC B cells and the T_FH_ cell subsets was investigated in our tissue cohort. To this end, follicles from all groups, harboring at least 20 cells for each corresponding cell population, were used. The X, Y coordinates of the relevant cells were extracted (Figure 2A, upper panel), and a digitalized representation of their distribution was generated (Figure 2A, lower panel). Then, the ‘G function’ parameter, as a surrogate for the dispersion/scattering of a distribution, of a given cell type, as well as the mean values of the minimum distances between relevant cell types in individual follicular areas, was calculated [34]. An example of follicles with a low and high mean distance between CD20^hi^ and PD1^hi^ cells and their associated G parameters is shown (Figure 2B). A similar distribution profile for CD20^hi^ cells among the tissue subgroups analyzed was detected (Figure 2C). Contrary to B cells, PD1^hi^ cells expressed a significantly higher dispersion in the vir-HIV subgroups compared to control LNs (Figure 2D). Regarding the PD1^hi^ cells distribution of cART tissues, a significant difference was found only between control and low-T_FH_ cART LNs (Figure 2D). Furthermore, a significantly higher dispersion was found in low-T_FH_ compared to high-T_FH_ HIV tissues, in both viremic and cART tissues from PLWH (Figure 2D). A significantly higher mean of minimum distance between CD20^hi^ and PD1^hi^ cells was found in the vir- and cART HIV compared to control tissues (Figure 2E), while no difference was observed among the PLWH subgroups (Figure 2E).

Then, the aforementioned parameters were calculated for the PD1^hi^CD57^lo^ and PD1^hi^CD57^hi^ cells in all tissue groups. Given the low abundancy of CD57^hi^ T_FH_ cells, a significantly lower number of follicular areas, especially in the low-T_FH_ HIV subgroups, was analyzed (Figure 2F,G). Again, a comparable CD20 G-function profile was found among the tissue groups in these follicular areas (Appendix A). Our data showed a similar G function and distance profile between tonsils and reactive control LNs (Figure 2F,G). In general, PD1^hi^CD57^lo^ and PD1^hi^CD57^hi^ cells express a significantly higher degree of dispersion in HIV-infected tissues compared to tonsils and control LNs (Figure 2F). Interestingly, the cART low-T_FH_ subgroup was the one with the highest dispersion of PD1^hi^CD57^lo^ cells among the PLWH subgroups (Figure 2F). With respect to the mean minimum distance, a trend, which was statistically significant for many of the comparisons, for a longer distance between CD20^hi^ and PD1^hi^CD57^lo OR hi^ cells, was measured in HIV-infected tissues compared to tonsils and control LNs (Figure 2G). Our data suggest a distinct T_FH_ cell in situ distribution profile, as well as a T_FH_-B cell proximity profile in PLWH, compared to tonsils and control LNs.

### 3.3. Significant Accumulation of Follicular Compared to Extrafollicular FOXP3^hi^CD4^hi^ T Cells in cART Low-T_FH_ LNs

We used the expression of FOXP3 as a surrogate for potential immune-regulatory CD4^hi^ T cells (Figure 3A). The Histo-cytometry gating scheme for the identification and calculation of FOXP3^hi^CD4^hi^ T cells in extrafollicular and intrafollicular areas (highly enriched in CD20^hi/dim^ cells) is shown (Figure 3B). The concordance between digitally/Histo-cytometry-identified FOXP3^hi^CD4^hi^ T cells (shown as spheres) and their original counterparts is demonstrated in Figure 3C. The lowest cell density of extrafollicular (EF) and follicular (F) FOXP3^hi^CD4^hi^ T cells was found in tonsils and the highest in reactive control LNs (Figure 3D,E). Comparable EF-FOXP3^hi^CD4^hi^ T cell densities among the PLWH subgroups were found (Figure 3D). Although not statistically significant, a trend for higher cell densities of F-FOXP3^hi^CD4^hi^ T cells in cART, compared to vir-HIV, LNs was measured (Figure 2E). A broad range of F-FOXP3^hi^CD4^hi^ T cell densities was observed, particularly in the cART low-T_FH_ subgroup (Figure 2E and Appendix A).

Next, the EF- and F-FOXP3^hi^CD4^hi^ T cell densities were compared in each LN. Similar EF- and F-FOXP3^hi^CD4^hi^ T cell densities were found in control LNs, while fewer F-FOXP3^hi^CD4^hi^ compared to EF-FOXP3^hi^CD4^hi^ T cells were found for almost all viremic PLWH tissues tested (Figure 3F). However, the opposite profile was observed for the cART tissues (Figure 3F). A consistent, significantly higher cell density of F-FOXP3^hi^CD4^hi^ compared to EF-FOXP3^hi^CD4^hi^ T cells was measured, especially in the cART low-T_FH_ subgroup (Figure 3F). The distribution profile (G function) of FOXP3^hi^CD4^hi^ T cells across the whole imaged area for each tissue was also investigated. No significant differences were found among the groups analyzed (Appendix A). Then, we focused our analysis on F-FOXP3^hi^CD4^hi^ T cells. An example of the identification and corresponding digital representation of their distribution in cART tissues is shown (Figure 3G). A significantly lower dispersion was measured in the cART low-T_FH_ compared to the cART high-T_FH_ subgroup, when the G factor was calculated for F-FOXP3^hi^CD4^hi^ T cells (Figure 3H). Our data show a preferential accumulation of follicular immune regulatory CD4^hi^ T cells in cART compared to viremic PLWH LNs.

### 3.4. LN GrzB^hi^CD8^hi^ T Cells Are Negatively Associated with Blood Viral Load

The in situ profile of bulk and effector (GrzB^hi^)CD8^hi^ T cells (Figure 4A) was investigated using a multiplex imaging assay, and the Histo-cytometry gating scheme is shown in Figure 4B. The applied antibody panel (Appendix A) does not include a follicular/GC biomarker; therefore, the cell density of the CD8^hi^ T cell subsets was analyzed for the whole imaged area (Figure 4A,B). The concordance between digitally/Histo-cytometry-identified GrzB^hi^CD8^hi^ T cells (shown as spheres) and their original counterparts is demonstrated in Figure 4C. In line with our previous data [12,15], an accumulation of bulk and GrzB^hi^CD8^hi^ T cells was measured in viremic donors, particularly the high-T_FH_ tissues, compared to tonsils and control LNs (Figure 4D and Appendix A). Although not significant, a reduction in GrzB^hi^CD8^hi^ T cells was observed between the viremic and cART tissues, which was more evident between the high-T_FH_ subgroups (Figure 4D). A positive association between circulating and LN bulk as well as GrzB^hi^CD8^hi^ T cells was found in HIV viremic samples (Appendix A). Contrary to circulating CD8^hi^ T cells (Appendix A), a significant negative correlation was observed between the viral load and the LN GrzB^hi^CD8^hi^ T cells (Figure 4E).

Next, the correlation between bulk and GrzB^hi^CD8^hi^ as well as FOXP3^hi^CD4^hi^ T cell densities was investigated. A significant association was observed between the two CD8^hi^ T cell populations in viremic LNs (Appendix A), as well as the cART low-T_FH_ subgroup (Figure 4F). However, this was not the case for the cART high-T_FH_ subgroup (Figure 4F). Among the groups analyzed, a positive correlation was found between bulk CD8^hi^ and EF-FOXP3^hi^CD4^hi^ T cell densities in the cART low-T_FH_ subgroup (Figure 4G, upper panel). This correlation was statistically significant between bulk CD8^hi^ T cells and F-FOXP3^hi^CD4^hi^ T cells in the same subgroup (Figure 4G, lower panel). The distance profiling revealed a similar dispersion of GrzB^hi^CD8^hi^ T cells in LNs from cART, compared to those from viremic PLWH (Figure 4H). Conclusively, our data revealed an accumulation of GrzB^hi^CD8^hi^ T cells in PWLH LNs compared to non-infected tissues, which was inversely correlated with the viral load in viremic PLWH.

### 3.5. Differential Modulation of Innate Immune Cell Subsets by cART

HIV is a chronic disease characterized by immune activation and inflammation [5,35,36]. Given the role of the innate immunity in HIV pathogenesis, we sought to investigate the tissue dynamics of several innate immune cell subsets using relevant biomarkers (CD163 and CD68, markers for monocytes/macrophages [37]; CD15, a surrogate for myeloid cells/granulocytes; CD16, a surrogate for activated myeloid cells, neutrophils and NK cells [38,39]) (Figure 5A). The Histo-cytometry gating scheme for the identification and quantification of these cell subsets is shown in Figure 5B. In general, a reduction was measured for all innate immune cell subsets analyzed in low-, compared to high-T_FH_ viremic, subgroups (Figure 5C), that was significant for CD15^hi^CD16^lo^ cells (Figure 5C). With respect to cART, lower cell densities of CD163^hi^CD68^lo^, CD15^hi^CD16^lo^, CD16^hi^CD15^lo^ and CD16^hi^CD15^hi^ cells were found in cART high-T_FH_ compared to viremic high-T_FH_ tissues (Figure 5C). Comparable cell densities of CD68^hi^CD163^lo^ cells were counted among all tissue groups analyzed (Figure 5C, upper panel).

Then, the relationship between CD8^hi^ T cells and innate immunity cell types was investigated. A positive association between the CD68^hi^ or CD163^hi^ cell subsets and bulk CD8^hi^ T cells in viremic LNs was found (Appendix A). A significant positive correlation between CD16^hi^CD15^lo^ and bulk or Grzb^hi^CD8^hi^ T cells was found in the vir-HIV high-T_FH_ group (Figure 5D). No such correlations were observed in cART LNs. Similar to Grzb^hi^CD8^hi^ T cells, a significant inverse correlation was also observed between CD16^hi^CD15^lo^ cells and peripheral blood viral load in viremic PLWH (Appendix A). An analysis of the in situ distribution profiling showed a higher dispersion in cART compared to viremic HIV-infected tissues for CD163 ^hi^ cells, while comparable profiles were found for CD68^hi^ and CD16^hi^ cells (Appendix A). Furthermore, a lower minimum mean distance between GrzB^hi^CD8^hi^ and CD16^hi^ cells, but not CD68^hi^ or CD163^hi^, in viremic compared to cART HIV tissues was found (Figure 5E and Appendix A). Altogether, our data suggest that the cell density of the individual innate cell types was differently affected in patients undergoing cART.

## 4. Discussion

Here, we have investigated the immune cell landscape in reactive LNs from PLWH and compared it to non-infected control LNs and tonsils (Appendix A). We chose to compare the in situ immune dynamics in HIV LNs to non-infected, reactive LNs characterized by active follicles (follicular hyperplasia), as a reference for highly active F/GCs. Given the abundance in GC of B and T_FH_ cells, as well as the preservation of the follicular and sub-follicular structures, tonsils are considered as a ‘prototype’ lymphoid organ for the investigation of the F/GC immune cell types. We should emphasize that none of the viremic PLWH had active opportunistic infections or AIDS-defining pathologies at the time of the sampling. The time since diagnosis was also similar among the individuals. Our analysis revealed two subgroups of HIV-infected LNs with respect to the cell density of T_FH_ cells, in both viremic and cART PLWH. This profile, at least for viremic individuals, is in line with the T_FH_ cell dynamics in the SIV non-human primate model [10,11].

No association of T_FH_ cell densities with gender, age, CD4^hi^ and CD8^hi^ counts was found in either viremic or cART PLWH. HIV infection of LN CD4^hi^ T cells per se, intrinsic T_FH_ cell factors and/or the interaction of T_FH_ cells with the GC microenvironment represent potential mechanisms that could contribute to the observed T_FH_ cell in situ dynamics by altering the differentiation and/or turnover rate of total (or specific subset) T_FH_ cells in PLWH. Regarding the viremic PLWH, we observed a negative association between pVL and GC/T_FH_ cells that could reflect a preferential infection/loss of T_FH_ cells or a generalized loss of T_FH_ cells in highly viremic PLWH. The relatively low numbers (~5% of T_FH_ cells) of HIV DNA+ T_FH_ cells [9] suggest that the HIV infection of T_FH_ cells per se may not be responsible for this negative association. Similar infection rates for CD57^hi^ and CD57^lo^ T_FH_ cells have been reported previously [40] challenging the preferential infection/loss of CD57^hi^ T_FH_ cells and the observed weaker association between these two T_FH_ cell subsets in viremic compared to cART PLWH. Abortive HIV infection represents an alternative mechanism for the loss of CD4 T cells, at least in vitro [41]. The role of such mechanisms in the regulation of vir-HIV T_FH_ cells needs to be investigated. Alternatively, progressive fibrosis and damage of vital LN structural elements (e.g., the Fibroblastic Reticular Cell network [42]) and/or the loss of GC survival signals (e.g., due to damage of the Follicular Dendritic Cell network [43]) for T_FH_ cells in high viremics could affect the differentiation and maintenance or turnover of T_FH_ cells. The observed heterogeneity of T_FH_ cell densities in different follicles across an individual tissue indicates that the locality of such mechanisms is possibly an important factor to consider in future studies. Our data urge for further investigation of LN structure elements, in conjunction with the in situ dynamics of immune cell types. The development of appropriate imaging tools will greatly facilitate such efforts.

Two groups (LoViReT and HiViReT) of treated PLWH, harboring a relatively wide range of very low viral reservoirs, was recently described [44]. Whether the cART T_FH_ subgroups correspond to a status is not known and merits further investigation. Despite viral control, cART is not able to fully restore the LN/follicular damage in PLWH back to normal. FRC reconstitution is one of the factors that could affect the reconstitution of the LN CD4^hi^ T cell pool and presumably T_FH_ cell prevalence [45]. No association of T_FH_ cell densities with treatment duration was found. However, the capacity of individual PLWH to differentially respond to cART and restore relevant LN elements could contribute to the observed high- and low-T_FH_ subgroups in the cART HIV group. Our distribution analysis showed an overall higher dispersion of PD1^hi^ T_FH_ cells in HIV-infected compared to non-infected tissues, which is also associated with a longer mean distance between T_FH_ and B cells in the infected donors. Previous studies have shown a reciprocal regulation between T_FH_ and GC B cells [46]. We hypothesize that the described spatial distribution profile may reflect a lower probability for these two cell subsets to interact in the infected LNs, leading to the subsequent loss of vital signals for T_FH_ cells. The aforementioned profile was more evident in the low-T_FH_ HIV subgroups, further supporting our hypothesis.

Follicular immune-regulatory CD4^hi^ T cells (T_FR_) represent an important ‘microenvironment cell factor’ for the development of T_FH_ cells [14,47,48]. Tonsils harbor the lowest number of FOXP3^hi^CD4^hi^ T cells, in line with previous reports [47,49]. Our data revealed a contrasting profile regarding the cell density between EF- and F-FOXP3^hi^CD4^hi^ T cells in control and viremic compared to cART-HIV LNs. Within the cART HIV group, we measured a significant increase in F-FOXP3^hi^CD4^hi^ T, specifically in the cART-low T_FH_ subgroup, suggesting a negative role for T_FH_ cell development in these individuals. Supplementary to this is the significantly less scattered distribution of FOXP3^hi^CD4^hi^ T cells within the follicles of low- compared to high-T_FH_ cART tissues. Whether the FOXP3^hi^ CD4 ^hi^ T cells represent bona fide T_FR_ cells or cells originating from T_FH_ cells [50] is not known and needs further investigation. Our data urge for a comprehensive in situ phenotypic and functional characterization of FOXP3^hi^CD4^hi^ T cells, especially in cART-HIV PLWH.

In contrast to PLWH, the majority of CD8^hi^ T cells in tonsils and control LNs express a GrzB^lo^ phenotype, in line with our previous observations [12]. The positive association between the numbers circulating and LN CD8^hi^ T cell density suggests that increased trafficking of bulk and presumably effector CD8^hi^ T cells may support, at least in part, their increased cell density in viremic LNs. A CXCR3/CXCR3L-mediated mechanism, which has been previously proposed for HIV PLWH [51] and SIV-infected non-human primates [15], could contribute to the CD8^hi^ T cell density profile we observed for viremic PLWH. Several studies have shown the role of CD8^hi^ T cells in controlling HIV and SIV [52,53]. In line with these studies, we found a significant negative correlation between LN GrzB^hi^ CD8^hi^ T cells and pVL that indicates a potential role of GrzB^hi^CD8^hi^ T cells in viral control, at least in part, in our cohort. An intermediate effector GrzB^lo^GrzK^hi^TOX^hi^TCF1^hi^CD39^hi^CD8^hi^ population negatively associated with plasma viremia was recently described in SIV/HIV-infected subjects [54]. One could hypothesize that different CD8^hi^ T cell subsets can contribute to viral control. The relative impact of individual CD8^hi^ subsets in viremic and cART PLWH is not known and needs further investigation. Therefore, the comprehensive in situ phenotypic (e.g., expression of homing receptors), functional (e.g., ‘regulatory’ function [55]) and spatial characterization of CD8^hi^ T cell subsets, especially in cART-HIV PLWH, is of great interest, given their potential role for immunotherapies aiming to eliminate the virus. We found a positive correlation between CD8^hi^ and F-FOXP3^hi^CD4^hi^ T cells, specifically in the low-T_FH_ cART-HIV subgroup. Whether and how these two cell types could affect the cell density of T_FH_ cells in this group remains to be elucidated.

Overall, we observed a diverse profile of innate immunity cell types among the groups. Individual cell types were differentially modulated by cART, while no consistent association with the low- or high-T_FH_ status was observed. We measured different cell densities for the CD68^hi^CD163^hi^ and CD68^lo^CD163^hi^ cell subsets. CD163 is a receptor that can be cleaved, and therefore, the cell density of CD163^hi^ macrophages/monocytes can be underestimated. Our results, however, indicate that the possible cleavage/loss of CD163 is not responsible for the observed in situ dynamics [44]. Our data point to a diverse innate immune LN microenvironment that could affect the host–virus interplay in these LNs [18]. We found a strong correlation between LN CD8^hi^ T cells and CD16^hi^CD15^lo^ cells in high-T_FH_ viremic tissues. This profile was also associated with (i) a less dispersed CD8^hi^ T cell distribution, (ii) a shorter distance between CD8^hi^ and CD16^hi^ cells and (iii) a negative association between pVL and LN CD8^hi^ or CD16^hi^CD15^lo^ cells. These findings urge for further investigation of neutrophils/granulocytes, in addition to macrophages, as possible key determinants for innate immunity/CD8^hi^ T cell crosstalk and virus dynamics [56]. Whether the same or different innate immune cell subsets mediate the host–virus interaction in viremic and cART PLWH is not known and remains to be elucidated.

Altogether, we have analyzed several immunological cell types in a well-characterized cohort of LNs and provide evidence (i) for the subgrouping of HIV-infected LNs (both from viremic and cART PLWH) based on their T_FH_ cell density, (ii) of a distinct profile of potential immunosuppressive FOXP3^hi^CD4^hi^ T cells in cART LNs with respect to their cell densities, distribution between extrafollicular and follicular areas and spatial distribution within the follicular compartment, and (iii) the effect of cART on the cell density of CD8^hi^ T and innate immune cells (Table 2). Altered T_FH_ cell density in HIV subgroups is associated with different CD8^hi^ and F-FOXP3^hi^CD4^hi^ T cell density and distribution profiles, too. The data suggest that further investigation of CD8^hi^ and immune-regulatory CD4^hi^ T cells could provide insights for the T_FH_ cell prevalence in HIV and particularly in cART PLWH. Understanding the follicular/GC microenvironment in HIV infection could further illuminate the role of T_FH_ cells in HIV pathogenesis and possible combinatorial interventions aiming to manipulate a major HIV tissue reservoir.

## Figures and Tables

**Figure 1 vaccines-12-00912-f001:**
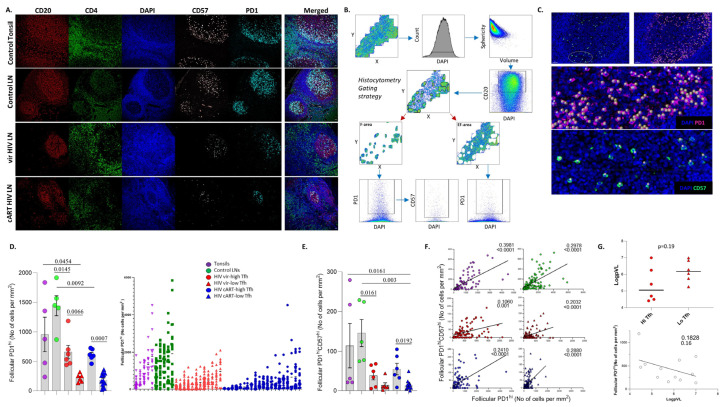
Similar in situ cell density of T_FH_ cells in viremic and cART PLWH LNs. (**A**) Representative examples of CD20 (red), CD4 (green), DAPI (blue), CD57 (gray) and PD1 (cyan) staining pattern from tonsil and control viremic and cART HIV LNs (scale bar: 30 μm). (**B**) The Histo-cytometry gating scheme used for the identification of T_FH_ cell subsets based on their expression of PD1 and CD57 is shown. F and EF areas were manually identified based on the density of the CD20 signal, and the relevant cell counts were extracted for specific tissue localities. (**C**) Histo-cytometry-identified PD1^hi^CD57^hi^ T_FH_ cells were backgated to the original image using Imaris software. Each sphere represents one cell. (**D**) Bar graph (left) demonstrating the cell density (normalized per mm^2^ counts) of follicular PD1^hi^ T_FH_ cells in tonsils (N = 5), control LNs (N = 5), viremic HIV (N = 12) and cART HIV LNs (N = 20). Viremic and cART were further subdivided based on their T_FH_ counts (HIV vir-high T_FH_ (N = 6), HIV vir-low-T_FH_ (N = 6), HIV cART-high-T_FH_ (N = 6) and HIV cART-low-T_FH_ (N = 14)). Each symbol represents one donor. The *p* values were calculated using the Mann–Whitney test and were corrected using FDR correction with q = 0.05 (Appendix A). Dot graph (right) shows the distribution of PD1^hi^ cell densities among the samples. Each symbol represents a follicle. (**E**) Bar graph demonstrating the normalized per mm^2^ numbers of follicular PD1^hi^CD57^hi^ T_FH_ cells in the same groups/subgroups of the samples. (**F**) Linear regression analysis to address the correlation between PD1^hi^ and PD1^hi^CD57^hi^ absolute follicular counts between different subgroups. Each symbol represents a follicle. R-squared and *p*-values are displayed on graphs. (**G**) Linear regression analysis (lower) to address the correlation between normalized follicular PD1^hi^ counts and blood viral load—pVL. Dot graph (upper) showing the pVL differences (as a log scale) between the viremic HIV subgroups.

**Figure 2 vaccines-12-00912-f002:**
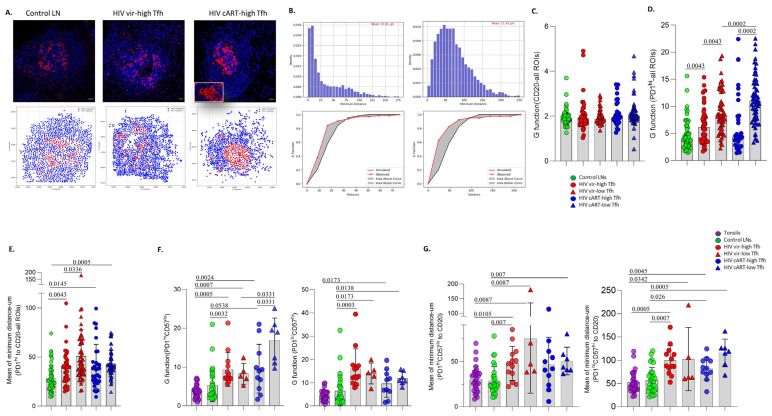
A highly scattered distribution of T_FH_ cells in HIV-infected compared to non-infected lymphoid tissues. (**A**) Representative immunofluorescence images showing the distribution of CD20^hi^ (blue) and PD1^hi^ (red) cells in follicles from one control, one vir-HIV high-T_FH_ and one cART HIV high-T_FH_ tissue (scale bar: 30 μm). The corresponding digitalized (generated by the Python distance analysis script) images are shown too. (**B**) The distribution bar graphs for the minimum distance between CD20^hi^ and PD1^hi^ cells in two follicular areas (control—left, cART—right) (upper panel). Diagrams showing the theoretical (blue) and experimental (red) Poisson curves for the distribution of PD1^hi^ cells in two follicular areas (control—left, cART—right) (lower panel). Bar graphs showing the G function analysis for total CD20^hi^ (**C**) and PD1^hi^ T_FH_ cells (**D**) in individual follicles from control and infected LNs (control LNs (N = 54), vir-HIV high-T_FH_ (N = 42), vir-HIV low-T_FH_ (N = 51), cART HIV high-T_FH_ (N = 31) and cART HIV low-T_FH_ (N = 55)). (**E**) The mean of minimum distance values between CD20^hi^ and PD1^hi^ cells in individual follicles from control and infected LNs is shown (control LNs (N = 54), vir-HIV high-T_FH_ (N = 42), vir-HIV low-T_FH_ (N = 51), cART HIV high-T_FH_ (N = 31) and cART HIV low-T_FH_ (N = 55)). (**F**) The G function analysis for PD1^hi^CD57^hi^ (left) and PD1^hi^CD57^lo^ (right) T_FH_ cells in tonsils, control and infected LNs is shown (tonsils (N = 29), control LNs (N = 29), vir-HIV high-T_FH_ (N = 13), vir-HIV low-T_FH_ (N = 5), cART HIV high-T_FH_ (N = 11) and cART HIV low-T_FH_ (N = 7)). (**G**) Bar graphs showing the mean of minimum distances between CD20^hi^ and PD1^hi^CD57^lo^ (left) or PD1^hi^CD57^hi^ (right) T_FH_ cells in tonsils, control and infected LNs (tonsils (N = 29), control LNs (N = 29), vir-HIV high-T_FH_ (N = 13), vir-HIV low-T_FH_ (N = 5), cART HIV high-T_FH_ (N = 11) and cART HIV low-T_FH_ (N = 7)). Each dot represents one follicle in all presented graphs. The *p* values were calculated using the Mann–Whitney test and were corrected using FDR correction with q = 0.05 (Appendix A).

**Figure 3 vaccines-12-00912-f003:**
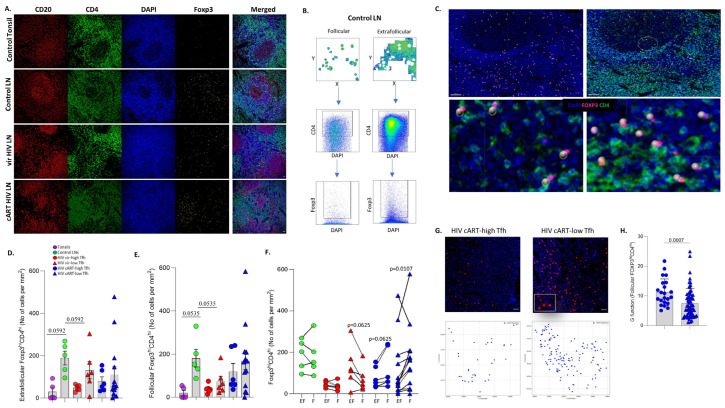
Preferential accumulation of FOXP3^hi^CD4^hi^ T cells in cART HIV follicular areas. (**A**) Representative examples of CD20 (red), CD4 (green), DAPI (blue) and FOXP3 (yellow) staining pattern from tonsil and control viremic and cART HIV LNs (scale bar: 30 μm). (**B**) Histo-cytometry immunophenotyping gating strategy used for the sequential identification of FOXP3^hi^CD4^hi^ T cells in follicular and extrafollicular areas in a control LN (**C**) Representative backgating of Histo-cytometry-identified FOXP3^hi^CD4^hi^ T cells, using Imaris software. Each sphere represents one cell. (**D**) Bar graph (left) demonstrating the normalised per mm^2^ counts of extrafollicular FOXP3^hi^CD4^hi^ T cells in tonsils (N = 5), control LNs (N = 5), HIV vir-high-T_FH_ (N = 6), HIV vir-low-T_FH_ (N = 6), HIV cART-high-T_FH_ (N = 6) and HIV cART-low-T_FH_ (N = 14). (**E**) Bar graph showing the normalised per mm^2^ counts of follicular FOXP3^hi^CD4^hi^ T cells in the same tissue samples. Each symbol represents one donor. (**F**) Bar graph with connecting lines demonstrating the normalized per mm^2^ counts of FOXP3^hi^CD4^hi^ T cells in follicular and extrafollicular regions in each tissue analyzed. (**G**) Representative immunofluorescence images showing the distribution of FOXP3^hi^ (green) and CD20^hi^ (red) cells in follicles from one cART-high-T_FH_ and one cART-low-T_FH_ tissue. The corresponding digitalized (generated by the Python distance analysis script) images are shown too. (**H**) Bar graph showing the calculated G function values for follicular FOXP3^hi^CD4^hi^ T cells in cART-high T_FH_ (blue circles) and cART-low T_FH_ (blue triangles) tissues (HIV cART-high T_FH_ (N = 22) and HIV cART-low T_FH_ (N = 64)). The *p* values were calculated using the Mann–Whitney test and were corrected using FDR correction with q = 0.05 (Appendix A).

**Figure 4 vaccines-12-00912-f004:**
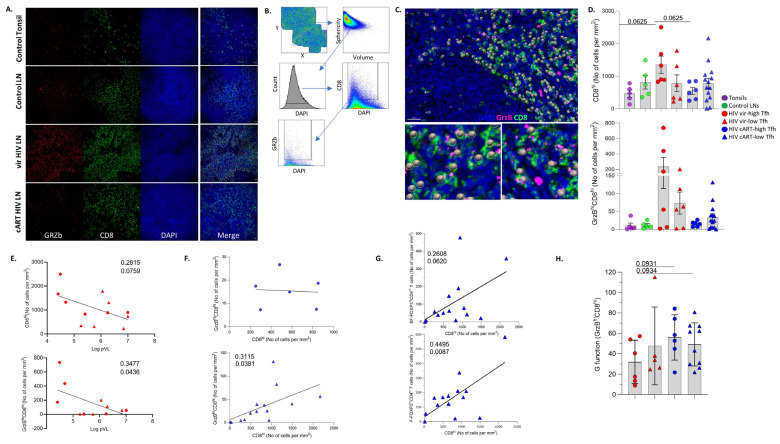
Accumulated LN GrzB^hi^CD8^hi^ T cells are negatively associated with PLWH blood viral load. (**A**) Representative examples of GRZb (red), CD8 (green) and DAPI (blue) staining pattern from tonsil and control vir HIV and cART HIV LNs (scale bar: 30 μm). (**B**) Histo-cytometry gating strategy used for the sequential identification of bulk CD8^hi^ and GrzB^hi^CD8^hi^ T cells. (**C**) Representative backgating of Histo-cytometry-identified GrzB^hi^CD8^hi^ cells using Imaris software. Each sphere represents one cell. (**D**) Bar graph (upper) demonstrating the normalized per mm^2^ counts of bulk CD8^hi^ cells in tonsils (N = 5), control LNs (N = 5), vir-HIV high-T_FH_ (N = 6), vir-HIV low-T_FH_ (N = 6), cART HIV high-T_FH_ (N = 6) and cART HIV low-T_FH_ (N = 14). Bar graph (lower) showing the normalized per mm^2^ counts of GrzB^hi^CD8^hi^ T cells in the same tissue samples. Each symbol represents one donor. (**E**) Linear regression analysis showing the correlation between LN CD8^hi^ (upper panel) and LN GrzB^hi^CD8^hi^ (lower panel) normalized T cell counts with blood viral load. (**F**) Linear regression analysis showing the correlation between LN CD8^hi^ and LN GrzB^hi^CD8^hi^ T cell counts in cART HIV high-T_FH_ (upper panel) and cART HIV low-T_FH_ (lower panel). (**G**) Linear regression analysis to address the correlation between extrafollicular (upper panel) and follicular (lower panel) FOXP3^hi^CD4^hi^ and LN CD8^hi^ normalized T cell counts in cART HIV low-T_FH_ subgroup. (**H**) Bar graph showing the G function values for LN GrzB^hi^CD8^hi^ T cells in the HIV subgroups (vir-HIV high-T_FH_ (N = 6), vir-HIV low-T_FH_ (N = 5), cART HIV high-T_FH_ (N = 6) and cART HIV low-T_FH_ (N = 10)). The *p* values were calculated using the Mann–Whitney test and corrected using FDR correction with q = 0.05 (Appendix A).

**Figure 5 vaccines-12-00912-f005:**
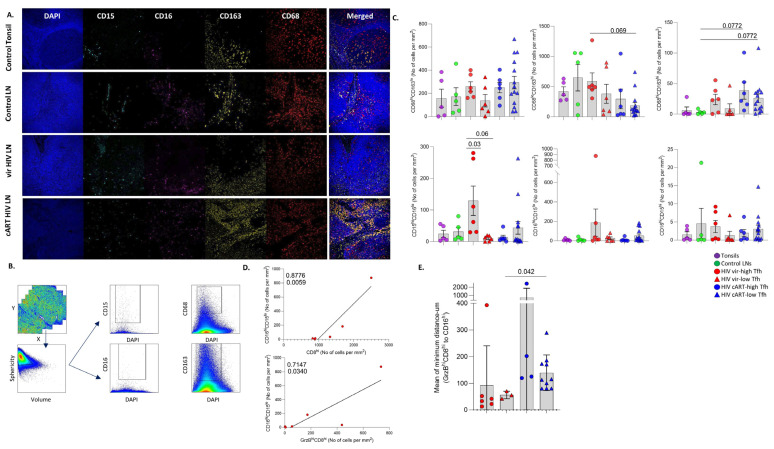
Altered innate immunity signatures among control and HIV samples. (**A**) Representative examples of DAPI (blue), CD15 (cyan), CD16 (magenta), CD163 (yellow) and CD68 (red) staining pattern from tonsil and control vir-HIV and cART-HIV LNs (scale bar: 30 μm). (**B**) Histo-cytometry gating strategy used for the sequential identification of CD15^hi^, CD16^hi^, CD163^hi^ and CD68^hi^ innate cells. (**C**) Bar graphs demonstrating the normalized per mm^2^ counts of bulk CD68^hi^CD163^lo^ (upper left), CD68^lo^CD163^hi^ (upper middle), CD68^hi^CD163^hi^ (upper right), CD15^hi^CD16^lo^ (lower left), CD15^hi^CD16^lo^ (lower middle) and CD15^hi^CD16^lo^ (lower right) in tonsils (N = 5), control LNs (N = 5), vir-HIV high-T_FH_ (N = 6), vir-HIV low-T_FH_ (N = 6), cART HIV high-T_FH_ (N = 6) and cART HIV low-T_FH_ (N = 14). Each symbol represents one donor. (**D**) Linear regression analysis to address the correlation between CD8^hi^ (upper panel) or GrzB^hi^CD8^hi^ (lower panel) T cells and CD16^hi^CD15^lo^ cells in vir-HIV high-T_FH_ tissues. (**E**) Bar graph showing the mean values of the minimum distances between GrzB^hi^CD8^hi^ T and CD16^hi^ cells in tissues from the HIV subgroups vir-HIV high-T_FH_ (N = 6), vir-HIV low-T_FH_ (N = 3), cART HIV high-T_FH_ (N = 4) and cART HIV low-T_FH_ (N = 10). Each dot represents a different donor. The *p* values were calculated using the Mann–Whitney test, and *p* values were corrected using FDR correction with q = 0.05 (Appendix A).

**Table 1 vaccines-12-00912-t001:** Demographic and clinical information of study PLWH.

ID (cART-HIV)	Age	Gender	Anatomic Location	CD4^hi^ Counts (Cells/μL)	Log pVL (Copies/mL)	Years HIV+	Treatment Duration (Years)	Current ART Regimen
cART-LN1	52	M	Inguinal	708	ND	6.5	5	EVG/COBI/3TC/TAF
cART-LN2	60	M	Inguinal	864	ND	22	22	DTG/DESCOVY
cART-LN3	56	F	Inguinal	437	ND	23	15	DTG/DESCOVY
cART-LN4	67	F	Inguinal	1168	ND	10	7.5	Biktarvy
cART-LN5	53	F	Inguinal	454	ND	26	11	Triumeq
cART-LN6	53	M	Inguinal	271	ND	9	14	DVC/COBI/TRV
cART-LN7	50	M	Inguinal	849	ND	32	15	Genvova
cART-LN8	59	M	Inguinal	587	ND	10	10	Atripla
cART-LN9	63	M	Inguinal	276	<40	14	4	RPV/DRV/RTV/DTG
cART-LN10	52	M	Inguinal	459	ND	13	1	ABC/3TC/DTG
cART-LN11	46	M	Inguinal	493	ND	3	3	EVT/COBI/3TC/TDF
cART-LN12	65	M	Inguinal	476	ND	26	25	FPV/RTV/3TC/TDF/MVC
cART-LN13	54	F	Inguinal	459	ND	13	1	ABC/3TC/DTG
cART-LN14	44	M	Inguinal	545	ND	16	7	ATV/RTV/FTC/TDF
cART-LN15	33	M	Inguinal	1121	ND	3y 5mo	2y 1mo	EVG/COBI/FTC/TDF
cART-LN16	51	M	Inguinal	805	ND	17	10	DTG/FTC/TDF
cART-LN17	37	M	Inguinal	662	ND	5	4	EFV/FTC/TDF
cART-LN18	53	M	Inguinal	441	ND	22	10	DTG/FTC/TDF
cART-LN19	57	M	Inguinal	500	<40	29	17	DRV/RTV/ABC/FTC/TDF
cART-LN20	58	M	Inguinal	636	ND	19	15	DRV/RTV/FTC/TDF
ID (Viremic-HIV)	Age	Gender	Anatomic Location	CD4^hi^ Counts (cells/μL)	CD8^hi^ Counts (cells/μL)	Treatment Duration (years)		Log pVL (copies/mL)
HIV-LN1	27	M	Cervical	520	175	No treatment		5.26
HIV-LN2	20	M	Cervical	290	881	No treatment		7
HIV-LN3	29	M	Inguinal	545	830	No treatment		5.41
HIV-LN4	22	M	Cervical	476	465	No treatment		5.75
HIV-LN5	37	M	Inguinal	391	1158	No treatment		6.25
HIV-LN6	24	M	Inguinal	453	3623	No treatment		6.84
HIV-LN7	29	M	Cervical	347	756	No treatment		7
HIV-LN8	33	M	Cervical	499	2211	No treatment		6.06
HIV-LN9	31	M	Inguinal	281	2142	No treatment		6.29
HIV-LN10	35	M	Cervical	491	2284	No treatment		4.49
HIV-LN11	27	M	Cervical	719	1831	No treatment		4.69
HIV-LN12	28	M	Cervical	777	2195	No treatment		4.49
ID (Reactive LNs)	Age	Gender	Anatomic Location					
LN1	74	M	Axillary					
LN2	41	M	Cervical					
LN3	21	F	Cervical					
LN4	49	M	Inguinal					
LN5	25	F	Inguinal					

**Table 2 vaccines-12-00912-t002:** A table summarizing the main comparisons, with respect to cell densities, for immune cell subsets among the tissue groups. The relative prevalence of a given cell subset is denoted as VL= very low, L = low, H = high and VH = very high, among the groups.

	PD1^hi^ T_FH_	PD1^hi^CD57^hi^ T_FH_	FOXP3^hi^CD4^hi^ (Follicular)	CD8^hi^ (Entire LN)	GrzB^hi^CD8^hi^ (Entire LN)	CD68^hi^CD163^lo^ (Entire LN)	CD68^lo^CD163^hi^ (Entire LN)	CD15^hi^CD16^lo^ (Entire LN)	CD15^lo^CD16^hi^ (Entire LN)
Tonsils	H	H	VL	L	VL	L	H	VL	VL
Control LNs	VH	H	H	L	VL	L	H	VL	VL
High-T_FH_ viremic	H	L	L	H	L	L	H	L	L
Low-T_FH_ viremic	L	L	L	L	L	L	H	VL	VL
High-T_FH_ cART	H	L	L	L	VL	L	L	VL	VL
Low-T_FH_ cART	L	L	H	L	VL	L	L	VL	VL

## Data Availability

The authors agree to share all publication-related data. Presented data are accessible through https://doi.org/10.5281/zenodo.12579627 (accessed on 25 September 2023) and used scripts are uploaded on https://github.com/LTI-CHUV/Distance-Analysis (accessed on 25 September 2023). For further information, please contact the corresponding author at konstantinos.petrovas@chuv.ch.

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
