# Peer review of "Follicular Immune Landscaping Reveals a Distinct Profile of FOXP3hiCD4hi T Cells in Treated Compared to Untreated HIV"

_vaccines, 2024, doi:10.3390/vaccines12080912_

Round 1

Reviewer 1 Report (Previous Reviewer 2)

Comments and Suggestions for Authors

The manuscript by Georgakis is much improved and with attention to the specific comments, will be acceptable.

Specific Comments

33-35. Reword for clarity: The significantly accumulated follicular, compared to extrafollicular, FOXP3hi CD4+ T cells found in the low-TFH cART-HIV group were characterized by a less scattered in situ distribution and strongly correlated with the cell density of CD8+ T cells in this group

60 Insert:    deregulated immune-regulatory (TREG) CD4 T cell and follicular immune-regulatory (TFR) CD4 T cell levels

194 What are the “original counterparts”? 

203 New paragraph?

268-269 Rephrase for clarity:  ‘….and neighboring with B cells…..

360 what does the CD8+T- mean?

391 hyphenate between bio- markers

413-414 Can this be stated more simply?  e.g cell density of the individual innate cell types was differently affected in patients undergoing  cART.

432 restate as:  immune cell landscape in reactive LNs from PLWH and compared it to non-infected control LNs and tonsils.  (no comma after non-infected)

436 restate as: GC of B and Tfh

437 add comma: structures, tonsils

445 restate as:  found in either viremic or cART….

451-452 What does this sentence mean?

456 I thought that the tonsillar T cells were a control and not infected with HIV

459 Where do the closed ) go?

500 restate: proposed for HIV……

501 no comma after [15]

505 the conclusion is not obvious and the statement is not necessary.

522 replace immunity with immune

Table 2. A legend is needed.  Define the ratings.  Comparisons of what?  Numbers, proximity, ? Also, is total tissue the entire lymph node or all the lymph nodes, both high and low ?

Why are there slashes in some boxes?

If GrzB CD8 is +++, then wouldn't the CD8 also be +++?

Author Response

Reviewer 2 Report (New Reviewer)

Comments and Suggestions for Authors

The manuscript by Spiros et al. investigates the follicular immune landscape of lymphoid nodes in people living with HIV (PLWH). Using advanced techniques such as multiplex immunohistochemistry (mIHC) combined with computational tools like histocytometry and neighboring analysis, the study presents several noteworthy observations: (1) The study highlights distinct profiles of follicular helper CD4+ T cell densities and distributions in PLWH compared to HIV-negative controls. These features remain consistent within groups, regardless of disease status, age, gender, CD4 count, ART status, or ART duration. (2) An expansion of FOXP3+ CD4+ T cells is observed in the ART group compared to untreated individuals, with these cells preferentially located in the germinal center (GC) area. (3) A negative correlation is found between plasma viral load and the densities of LN GrzBhiCD8+ T cells and CD16hiCD15lo (likely NK) cells, indicating their potential antiviral activity.

While the study is well-introduced and thoroughly discussed, several areas require better organization and clarification. Below are specific concerns and recommendations.

(1) Patients were categorized into high-TFH and low-TFH subgroups based on TFH cell densities. However, the significance of this grouping is unclear, as most meaningful observations are seen between treated (ART) and untreated (TN) groups. Given the limited sample size, it is not evident that further subdivision into high-TFH and low-TFH subgroups is necessary for correlation analysis.

(2) Reference 12 appears to be incorrectly cited and does not relate to the mentioned content. This needs to be rectified for accuracy.

(3) Different lymphoid tissue samples were obtained from various institutions. The manuscript should provide detailed information on how sample preparation consistency was maintained across different sites to ensure the reliability of the results.

(4) The manuscript should include the R values for the correlation analyses to quantify the strength of the observed relationships.

(5) In Figure 4E (upper panel), all data points are distributed on one side of the trend line and are distant from it. This raises concerns about data accuracy. The authors should verify and confirm these data points.

(6) It would be beneficial to discuss whether LN GrzBhiCD8+ T cells share any features with the recently reported TOXhiTCF1+CD39+ CD8+ T cells (Nat Immunol. 2024;25(7):1245-1256).

(7) In Figure 5c, markers are expressed as+/-, whereas text uses notation hi/lo. This should be standardized for clarity.

(8) The manuscript should include a table comparing patient groups, providing a clear overview of the differences and similarities.

(9) Some items in Table 1 lack units, such as Treatment Duration and log pVL. Units should be provided for all metrics to ensure clarity.

Author Response

This manuscript is a resubmission of an earlier submission. The following is a list of the peer review reports and author responses from that submission.

Round 1

Reviewer 1 Report

Comments and Suggestions for Authors

The authors performed multiplex staining of LNs from viremic and antiretroviral treated people living with HIV, and reactive, non-infected control LNs and tonsils in order to elucidate follicular (F)/germinal center (GC) immune landscape. Here are some suggestions for improvement of manuscript and questions for clarification.  

·       What is authors opinion, weather different geographical origin of  samples of viremic LNs (Mexico City) and cART HIV LNs (Seattle) might influence the results, regarding types/subtypes of HIV virus?

·       From which anatomical sites of patients are taken LNs for analysis?

·       Based on which criteria authors defined follicular area? Are there differences in their surfaces or surfaces of GC?

·       Why marker CXCR5 was not included in analysis of Tfh cells?

·       Including H&E staining of the analysed tissues in manuscript would give added value to the paper.  

·       Throughout the paper, designation of cells based on expression or no expression of some marker should be with “+” or “-“. For example CD4+ T cells, CD8+ cells…..

·       Rewrite the sentence (lines 245 - 247) “Therefore, the differential expression of TFH cells in PLWH LNs is also associated with a distinct profile of in situ distribution and neighboring with B cells”. Cells can not be expressed.

·       Full name for abbreviation PLWH should be written (line 46).

·       Equalize marking of Tfh cells (TFH or Tfh) (line 82).

Reviewer 2 Report

Comments and Suggestions for Authors

Georgakis et al describe an immunofluorescence study of lymph nodes from normal, HIV infected, and  HIV infected and treated individuals highlighting the presence and proximity of CD4 Tfh; CD8 T cells, with and without high levels of granzyme; and  other cells.  The images and their analysis demonstrate that  there are fewer Tfh in HIV infected and HIV treated individuals (fig 1), this also means that the distance between Tfh and B cells is greater with HIV infection (fig 2), Control lymph nodes contain more T regs than HIV infected but if higher Tfh, then lower Treg (fig 3),  CD8 T cell numbers are increased with HIV infection and effector (granzyme positive) cells decrease with treatment (fig 4), and finally,  macrophages, innate lymphoid cells (including NK) and other innate cells are most prevalent in HIV treated  lymph nodes.   This reviewer arrived at these conclusions from viewing the figures and not from the text, which was difficult to follow. 

The conclusions from the findings support the subtitle but not the title of this manuscript.  Proximity and numbers do not reveal a role only a presence for these cells in which they may have been recruited for other reasons. The simplest explanation for this study is that the greater distance between cells is fewer numbers within a lymph node of a particular cell type rather than a conclusion about function.  What other reason might there be? A more accurate title is necessary.  Specific comments are below:

It is not clear whether the Tfh high and Tfh low lymph nodes are from the same person or different people.  If different people, then there can be many reasons why there are differences in the parameters that were studied. 

The source of the reactive LNs characterized by follicular hyperplasia is not indicated.  Also, figure S1 does not indicate the source of the LNs.  Also, are these the control LNs used throughout the manuscript?

Line 31 (PLWH)and

32, 35 define high Tfh and low Tfh more clearly if using in abstract.

46 All abbreviations must be defined within the manuscript.  PLWH

53. Rephrase since cells are not a site.

54 Where are the Tfh cells accumulating?

56 This sentence contradicts previous sentence unless previous sentence is modified, hence confusion.

80  GrzBhiCD8Tand

81-83 rephrase for clarity. 

158 define pVL

177 Does this mean that Tfh cells express low levels of CD4? Do other Tfh cells express more CD4?  Compromised implies inhibition.  It could be that these are the survivors of HIV infection but we don’t know.

181 Did you mean to use TFH or other abbrev.  Why is the PD1 expression for follicular CD8 T cells significant to this section?

186  New thought, new paragraph

198 Where did this comment come from?  The straightforward observation of the figure is a deficit of Tfh in HIV LNs rather than what is stated.   

245 The straightforward conclusion is that greater distance between these cells correlates with fewer Tfh in HIV infected LNs.

280 New thought, new paragraph

286 rephrase sentence, “That was more evident for the 286 cART low TFH subgroup (Figure 3F).”

 319 Unclear whether ‘non-infected tissues’ refers to LNs, tonsils or liver and whether these are from control patients or from HIV patients that had unaffected tissues.  

332 What is a viremic tissue?  Which tissues?

332-333 How can this statement be made?  Is this conclusion based on numbers or proximity?  This reviewer concludes that CD8 T cells decrease with treatment as shown for HIV panel A and D.  One may conclude that with treatment, fewer effector CD8 T cells may be necessary.

390.  A table(s) that compare the numbers and distances indicated by ++ rather than numbers would be useful to summarize the key findings of this study. The discussion is hard to read and would benefit from a summary of the take home lessons. Where conclusions are made, there needs to be better justification.  The simplest explanation for this study is that the greater distance between cells is fewer numbers within a lymph node of a particular cell type rather than a conclusion about function.  What other reason might there be?

394. Replace Despite with Because

398  This statement should be qualified with regard to whether the lymph nodes (Tfh low and Tfh high LNs) were from different people?  If different people, then there could be other reasons for the differences.

401 Rephrase: Given that tissues from both viremic and cART individuals….

405 What does “directional differentiation” mean?

406 Do you mean that most PD1hi cells were also  CD57hi?

412 What is the basis and consequence for this hypothesis?

414 PLWH.A

428 Meaning of CD4 T selectivity ?

434 Replace Contrary with In contrast to

438 is CD8/GrzB…. a ratio?  If so, please spell it out.

445-448 Rephrase for clarity

451 Why make the suggestion re: Tfh cell development when they may just not be recruited or killed off.

452 Discuss the CD163 cell data first and the caveat in this sentence after as rationaliztion.

462  Why can't proximity mean that CD8 T cells are activating or controlling the innate cells rather than the other way around?  Your conclusion needs justification.

467 As stated in the beginning, this summary is insufficient and should be more of a list of the major conclusions of the study.  The text above does not summarize. 

Comments on the Quality of English Language

English is readable but word choice and phraseology is problematic is certain places.

Round 2

Reviewer 2 Report

Comments and Suggestions for Authors

This manuscript contains an interesting analysis of the presence and position of Tfh and other cells in lymph nodes from viremic and treated HIV patients compared to normal lymph nodes.  In as such, there could be interesting information from this manuscript.  However, despite the modifications of the authors, the manuscript remains very difficult to read and filled with unsubstantiated statements. Each of the subsections within the results section is based on a figure and as such should first describe simply what is in the figure starting with the obvious and then progressing through the more sophisticated analysis. Help the reader see what you see.  For example, in Figure 1, the distribution of CD20 cells should be described to define the GC and the differences for the different populations.  As a result, the authors jump to the T cell descriptions before defining the GC within their images (more about this figure in specific comments).  

The subsections, based on the figures,  should not end with a subjective and speculative comment but a true summary of what is learned from these data, the take home lesson. For example, for Figure 1, a summary of the distinction for high and low Tfh.  Also, was there any possible reason or differences between these lymph nodes to cause the distinction.   Speculation and subjective statements should be retained for the discussion, with appropriate rationalization. 

The authors indicated that a summary table would be inserted into the manuscript but no such table was in this revision.

This study has the potential to be interesting but the manuscript must be totally rewritten before it will be acceptable.  It requires more revision than just addressing the specific comments that follow.

Specific Comments.

60 define Tfr

80 define Grz

140 reword:.”.......relevant cell subsets (*****) cells was.....” for clarity

144 define ROI

147 do you mean NumPy  ?

174 In looking over Fig 1A and S6, it is hard to see a GC in the HIV infected  LNs.  Unlike uninfected LNs, there is no defined structure to the CD20 cells. This is the most obvious finding and should be stated.  As such, what does this say about GCs in HIV infected individuals?  Fig 2A looks more like a GC for the HIV LNs. 

174 What is a high-Tfh subgroup?  The Low and high Tfh groupings need to be clearly delineated and described as they become a major part of the discussion.

192  Define the strong association (numbers of cells, closeness, coexpression of hi levels of  PD1 and CD57 on the same cells)?

196 what is ‘vir-low’

198-199 Suggested rephrase:  two subgroups defined by significantly different cell densities of Tfh cells were identified for viremic as well as cART HIV LNs.

245-247 I don't see a big difference for the PLWH LNs compared to untreated and these are very different from uninfected.

272 what does this mean:  We used the expression FOXP3 as a surrogate.     ??expression OF FOXP3??

276 What does “balanced expression” mean?

277 I don't see FOXP3 in the untreated LNs in 3A.

293-294  This conclusion is not directly from the data.  Better conclusion:  FOXP3 cells dissipate with HIV but increases with treatment.  There is no connection in this figure to connect Tfh and FOXp3 cells

333-336  No conclusion or even suggestion can be made about the  function of the cells based on the data.  The best that can be said is that there is a correlation between CD8 T cell presence and HIV status.  Suggestions should be reserved for the discussion, but only if backed by the data.

From Fig 4a, best conclusion is that HIV infection draws grz+CD8T cells into LN with more of these cells during active infection than therapy controlled infection.

316 do you mean preformed or performed?

364-366 remove the commas

375-377 The  data points to cell proximity but says nothing about a network.

403 The cell density of Tfh in LNs deserves more discussion and its significance for the different groups.

406-408 No basis for this conclusion was presented in the data or poorly described.

409-411 What is the data basis for the hypothesis?

421-424. Need citation for role of Tfh for viral reservoir.  The rest of this statement is very amorphous with no follow up.

448-449  This statement is a stretch and needs to be rephrased with better arguments.

Comments on the Quality of English Language

English is ok